# Enhancing Vietnamese Students’ Acceptance of School Lunches Through Food Combination: A Cross-Over Study

**DOI:** 10.3390/nu17081385

**Published:** 2025-04-20

**Authors:** An Thanh Truong, Anh Thi Lan Pham, Thy Quynh Nguyen, Tan Duy Doan, Tuan Nhat Pham, Yen Thi Hai Hoang, Ryosuke Matsuoka, Shigeru Yamamoto

**Affiliations:** 1Faculty of Public Health, University of Medicine and Pharmacy at Ho Chi Minh City, Ho Chi Minh City 72760, Vietnam; lanh2804@gmail.com (A.T.L.P.); nqthy234@gmail.com (T.Q.N.); doanduytaan@ump.edu.vn (T.D.D.); phamnhattuan@ump.edu.vn (T.N.P.); 2Asian Nutrition and Food Culture Research Center, Jumonji University, Niiza 352-8510, Japan; hoanghaiyen06112000@gmail.com (Y.T.H.H.); yamamotoshigeru426@gmail.com (S.Y.); 3R&D Division, Kewpie Corporation, Tokyo 182-0002, Japan; ryosuke_matsuoka@kewpie.co.jp

**Keywords:** school lunch, school meals, Vietnamese, vegetable intake, cross-over study

## Abstract

Background/Objectives: Vegetable leftovers constitute more than half of Vietnamese school lunch waste, partly due to limited ingredient variety, which may reduce meal acceptance. Methods: This cross-over study assessed the impact of diversifying vegetable options on intake and satisfaction among 40 students at a suburban Vietnamese primary school. Five new menus were developed by redistributing a 100 g vegetable portion into smaller servings of multiple vegetable types, combining them creatively with protein-rich foods or rice while maintaining nutritional value and cost. Students alternated between current and new menus over four weeks. Sensory evaluations using a 5-point hedonic scale and food weighing were conducted daily. Results: Most students increased vegetable intake during the new menu period. Mean intake was significantly higher with new menus (81.5 g; 95%CI: 77.1–85.9) compared to current menus (71.1 g; 95%CI: 65.2–75.1) (*p* < 0.001). Conclusions: These findings demonstrate that enhancing vegetable variety in combination can significantly improve intake and sensory characteristics without additional costs. This scalable strategy offers a practical solution for schools to foster healthier eating habits among students.

## 1. Introduction

Vietnamese school lunch programs have been implemented since 1980, with the first purpose being to provide lunch meals for students [1,2]. From 2022 onwards, the Vietnamese Ministry of Education and Training promulgated a “Guideline on organizing school meals combined with increasing physical activity for children and students in preschool and primary education establishments”. This guideline emphasizes school lunch as a strategy to improve Vietnamese stature and health [3]. On the other hand, as eating preferences often persist from childhood into adulthood, school lunch meals can be essential to promote healthy eating habits [4].

Despite these intentions, school lunches in Vietnam face significant challenges. School lunches contribute 30–40% of students’ daily needs [2,3]. However, 23% of served meals (approximately 85 g of food) are estimated to be left uneaten. The prevalence of sharing of total leftovers according to rice dishes, protein dishes, and vegetable dishes are 15%, 27%, and 58%, respectively [5]. Large amounts of leftovers may prevent students from receiving full nutritional benefits from school lunches. This amount of leftovers also shows an urgent need to improve Vietnamese school lunches.

Although there is a regulation requiring least ten food ingredients in each meal, Vietnamese school meals do not meet this criterion [6,7]. According to a study comparing school lunches in Vietnam and Japan, both countries have 190 school meals in a year. While Vietnam cycled 23 menus throughout 190 meals with 53 food ingredients, there were 190 different meals with 376 food ingredients in Japan [8]. There are no school dietitians in Vietnam, so the vice principal or kitchen staff mainly prepares menus without sufficient nutrition or knowledge of children’s health. This leads to a limited variety and high repetition of dishes [1,5,8]. Since eating various foods has been demonstrated to improve food intake, this issue may be the main reason for the large number of leftovers in Vietnamese school lunches [9].

A combination of foods in a meal can increase the diversity and the meal’s color, which has been scientifically proven effective in improving food intake, especially vegetables [10,11,12]. Japanese school meals usually use a mixture of vegetables, vegetables and meats, or vegetables and rice, increasing the appeal to encourage students to finish their meals [8,13]. Unlike the Japanese, Vietnamese cook all ingredients separately in a simple way, with all vegetables served as stir-fried or soup [5,13]. This monotonous cooking cannot make the meal colorful enough to increase the students’ appetite and make them unwilling to eat all the meals served [5,14]. Thus, Vietnamese school lunch’s vegetable dishes leftover was reported higher than Japanese with 58% and 20%, respectively [5,15].

Some strategies have been implemented to decrease Vietnamese school lunch leftovers. However, they were all educational programs that educated students about the waste of food and the nutrient benefits [16,17]. These programs successfully increased the students’ knowledge but not their attitudes or behavior. Consequently, although their understanding did increase, their consumption of vegetables or food did not [16,17]. On the other hand, Martins et al. (2020) and Zhao et al. (2019) came to the same suggestion that the sensory characteristics of school lunches needed to improve to reduce the food left uneaten [14,18]. From these points, changing school meal menus seems to be a promising approach to gradually boosting students’ acceptance of food and vegetables.

Based on the above, this study was conducted to test the hypothesis that food combinations in school lunch meals can positively enhance Vietnamese students’ acceptance of school lunches focusing on vegetables.

## 2. Materials and Methods

### 2.1. Study Design

A cross-over design was undertaken at a suburban public primary school in Ho Chi Minh City from March to April 2024. Participants were fifth-grade students (10–11 years old) who participated in the school lunch program, were healthy and had no food allergies.

### 2.2. Sample Size Calculation


n≥Z1−α2+Z1−β2(σ12+σ22r)(μ1−μ2)2


The sample size was estimated using the formula for calculating a difference with the previous mean and standard deviation with a 95% confidence level, sample ratio in the two groups r = 1, the average amount of vegetables consumed between two groups µ_1_ = 54.7 g, µ_2_ = 79 g, standard deviation σ_1_ = 25.0 g, σ_2_ = 16.3 g (based on research evaluating the impact of the new menu on the amount of vegetables consumed by students aged 10–11 in Hanoi, by author Nguyen Van Diep) [7]. The sample size was determined using WHO Sample Size 2.0. The minimum required sample size was 16 subjects per group. In fact, we recruited a total of 20 subjects per group.

### 2.3. Study Procedure

#### 2.3.1. Developing New Menus

We developed new menus based on the school’s current menus. First, we collected current menus for 1 week and all the ingredients allowed for school lunch meals. We then made ten new menus by redistributing the 100 g portion of one or two vegetables from the current menu into smaller servings of multiple types of vegetables, aiming to increase variety and appeal while maintaining the same nutritional content and meal cost. We kept the same seasonings as the current menu and followed all the guidelines of The Vietnamese Ministry of Education and Training [6]. After that, a randomization of 10 students were invited to taste and give a score from 1 to 5 for 10 menus. Finally, we selected the top 5 menus from the highest to lowest scores for the main study. The final new menu developments are shown in Appendix A.

#### 2.3.2. Intervention Period

The school currently uses a 3-week cycle menu, so we conducted our study on the first week and fourth week of the month to ensure that the current menu was the same between the two groups.

One group (20 students) consumed the current menu in the first week and switched to the new menu in the fourth week, while the other group followed the reverse order. Study procedures are shown in Figure 1.

All the students in two groups were invited to have meals in two different rooms, separate from the casual dining area of the school.

### 2.4. Data Collection and Analysis

#### 2.4.1. Data Collection

We weighed all the food before and after cooking. Then, we numbered the trays by the subjects’ ID and served them to the subjects. After meals, we collected the tray, weighed each leftover component separately, and recorded the data based on the number of subjects. We weighed all the food using the same food scale, Tanita KD-160, with an accuracy of 0.1 g daily.

The original menus included a watery soup every meal. From our observation and previous data, in Vietnamese meals, students often mix the watery soup with the rice to easily swallow the remaining foods [17]. Some uneaten rice became wet in the soup and became heavier when weighed. In these cases, we only removed the soup water using a small sieve and weighed the rice (or vegetables, meat). Thus, 15% may overestimate the amount wasted on both types of menus [5].

Sensory tests were conducted using a 5-point Hedonic scale, following the questionnaire of author Nguyen Van Diep, with these five questions below [7].

How do you rate the color of this meal?How do you rate the taste of this meal?How do you rate the smell of this meal?How do you rate the texture (uniform surface, crunchiness) of this meal?Overall, how do you rate this meal?

Students then gave scores of 1 (dislike very much), 2 (dislike), 3 (neither dislike nor like), 4 (like), and 5 (like very much) for each question.

#### 2.4.2. Data Analysis

Energy, protein, lipid, and carbohydrate levels were calculated using the Vietnamese Food Composition Table 2017.

Quantitative variables were checked for normal distribution and shown as mean ± SD for variables with normal distribution or median (inter-quartile range) for variables with abnormal distribution. Comparisons of each individual intake amount between the new menu and the current menu period were performed using Student’s paired *t*-test or the Wilcoxon signed-rank test. *p*-values of less than 0.05 were considered statistically different for all the analyses. The above statistical procedures were performed using Stata version 17.0 (StataCorp. 2021. Stata: Release 17. Statistical Software, StataCorp LLC, College Station, TX, USA).

### 2.5. Ethics Approval and Consent to Participate

Procedures for this study were followed according to the ethical standards from the Helsinki Declaration and were approved by the Biomedical Research Ethics Committees at the University of Medicine and Pharmacy at Ho Chi Minh City, Ho Chi Minh City, Vietnam (number 329/HĐĐĐ-ĐHYD, 2024). Written informed consent was obtained from all the subjects’ caregivers.

## 3. Results

This study was conducted on 40 fifth-grade students at a public primary school in suburban Ho Chi Minh City, Vietnam.

### 3.1. Menu Cost and Nutrient

Table 1 compares the provided meals between five current menus and five new menus. Although the cost was the same between current and new menus, food and vegetable varieties in new menus increased 1.6 and 3.1 times compared to current menus (*p* < 0.001).

### 3.2. Sensory Test

Table 2 demonstrates the average score of sensory tests for five current menus and five new menus. Although the sensory test scores for the current menus were already high, they were statistically improved with the new menus (*p* < 0.05).

### 3.3. Vegetable Intake

Figure 2 illustrates the comparison of vegetable intake between current and new menus. The vegetable intake on the current menu was 71.1 g (95%CI: 65.2–75.1), significantly lower than that on the new menu with 81.5 g (95%CI: 77.1–85.9) (*p* < 0.001). The average difference between the two types of menus was 10.4 g.

Figure 3 illustrates that the majority of subjects—approximately 90%—consumed more vegetables with the new menus compared to the current ones. The increase in vegetable intake varied among subjects. While the most substantial improvement reached up to 40 g, indicating a significant dietary change for some individuals, others showed only modest increases or no change at all. A few subjects even showed minimal or no improvement in their intake. Notably, subjects with initially low consumption on the current menus were able to boost their intake on the new menus significantly. Additionally, more subjects consumed the whole vegetable portion under the new menus, in contrast to just one subject who did so under the current menu. These findings suggest that individual responses varied while the new menus were broadly effective.

### 3.4. Energy and Nutrient Intakes

Table 3 indicates that not only did vegetable intake increase, but energy, protein, lipid, and carbohydrate intake also increased significantly compared to the current menu results within the same amount of supply between the two menu types. All the differences were statistically significant, with *p*-values less than 0.001 (energy, lipid, carbohydrate) and *p*-value less than 0.05 (protein intake).

## 4. Discussion

This study investigated students’ acceptance of school meals by providing a variety of vegetables to create new combinations. Based on the sensory test, students evaluated higher scores in all components, including color, taste, smell, texture, and overall, with the updated menus. Additionally, there was an increase in the intake of vegetables and meat (or fish) during the new menu phase. These findings indicate that combining vegetables with other vegetables, meat (or fish), or rice enhances sensory scores and increases food intake while maintaining meal costs. This supports the proposed strategy to improve school lunch meals.

Although a 10 g increase in vegetable intake per meal may seem small, it is clinically meaningful for Vietnamese children aged 10–11, particularly in light of national concerns about both undernutrition and the rising prevalence of overweight and obesity [19]. Current data indicate that vegetable intake among Vietnamese school-aged children remains below recommended levels [20]. Therefore, small, achievable dietary changes, such as a 10 g increase per meal, resulting in an additional 30 g per day, are more likely to be adopted and sustained, especially in school settings. This incremental improvement can help move children’s intake closer to the dietary guidelines set by the Vietnamese Ministry of Health and the National Institute of Nutrition (Available in Vietnamese at: https://moh.gov.vn/tin-noi-bat/-/asset_publisher/3Yst7YhbkA5j/content/bo-y-te-cong-bo-ket-qua-tong-ieu-tra-dinh-duong-nam-2019-2020, accessed on 15 February 2025).

Based on the sensory test scores, our study suggests that combining food ingredients had a positive impact on the meals’ sensory qualities. Attributes such as appearance, smell, and texture are important predictors of whether children choose to eat a meal [21,22,23]. Francesca Zampollo emphasized that children tend to prefer plates that include a variety of items and vibrant colors [24]. Similarly, Nguyen Van Diep reported a 24.3 g increase in vegetable intake among students when meals were enhanced in sensory test scores, specifically in terms of color, smell, taste, and texture [7]. Aligned with these findings, our results showed that the new menus, which scored higher in color, smell, and texture, were associated with lower quantities of vegetables left uneaten compared to the current menus.

Palatable cuisines worldwide are characterized by high levels of umami-related substances, including glutamate, inosinate, and guanylate [25,26,27,28]. Studies have demonstrated that combining glutamate with inosinate or guanylate produces a synergistic effect, significantly enhancing flavor perception [29,30]. This phenomenon, known as the “synergistic effect of taste”, provides a scientific basis for flavor enhancement [29]. Naturally, these substances are found in different food sources: glutamate is abundant in vegetables, inosinate in protein-rich foods such as meat and fish, and guanylate in mushrooms [31,32]. Thus, pairing vegetables with meat, fish, or mushrooms can substantially boost flavor. This mechanism likely explains the observed increase in taste intensity among subjects, as the average taste score improved significantly from 4.5 in the current menus to 4.8 in the new menus (*p* = 0.01). Given that many studies have identified good taste as a crucial factor influencing children’s food acceptance [21,33,34], our study’s enhanced taste scores could have significantly contributed to increased intake of all meal components.

In our study, several high-glutamate vegetables permitted in the food ingredients list, such as tomatoes and cabbage [35], facilitated the development of menus leveraging the synergistic taste effect. Observational data indicated that students responded most positively to the meal served on Wednesday (Appendix A). In this menu, the original ketchup to eat with a fish ball was replaced with a tomato-based sauce containing diced carrots and onions, creating a combination of glutamate (tomato, carrot, and onion) and inosinate (fish) that appeared to enhance palatability among participants. Similarly, on Friday, the mixed rice dish (comprising rice, vegetables, egg, and pork) and vegetable-only side dish (cabbage, tomato, and cucumber) were offered (Appendix A). Although no statistical analysis was conducted, anecdotal observations suggested that students were more likely to consume vegetables when incorporated into the mixed rice (including protein-based ingredients), while vegetable-only dishes were left uneaten with larger amounts. Unfortunately, mushrooms were not included in the study. According to the vice principal, purchasing mushrooms in small quantities was deemed cost-prohibitive, potentially increasing the overall cost of the new menus. Therefore, future research is recommended to include a broader sample of subjects to examine the effects of incorporating mushrooms into combinations of vegetables on vegetable consumption and menu acceptability.

The preparation method also influenced vegetable consumption among children [36]. Research specifically examining the vegetable preparation preferences of children 10 to 11 is limited. However, Zeinstra et al. (2010) [36] concluded that all age groups in their study, including 11–12-year-old children, preferred steamed and boiled vegetables over mashed, stir-fried, grilled, and deep-fried, with the most familiar preparation belonging to boil. Crunchiness and color also proved to be characteristics of vegetable preferences among children. Thus, shorter boiling times for vegetables might result in retaining crunchiness and avoiding undesirable brown coloring in stir-fries, enhancing children’s willingness to finish their vegetable dishes [36]. Regardless of the above, from our observation, vegetable cooking methods were limited in Vietnamese school meals. A previous study reported that all vegetables served in primary schools’ lunches were stir-fried or included in soup, but no boiled vegetables or salads were served [5]. Food providers and school staff have previously deemed boiled vegetables impractical due to the difficulty of serving sauces, and fresh vegetables are not permitted in Vietnamese school meals [5]. Nevertheless, the “boiled salad” seemed effective in our study, and this method has been widely used in the Japanese school meal program [13]. We served a dish of boiled vegetables, comprising cabbage, carrot, broccoli, and corn, then mixed well with sesame dressing before dividing each subject’s portion (we committed to following all the rules of the school lunch program in Vietnam and the balance of nutrient was no difference compared to the current one on the same day). Although we did not statistically analyze the effectiveness of this preparation method, observational data suggested that most students enjoyed these boiled vegetables more than the other options. Further studies could explore the impact of this method on children’s vegetable consumption.

Some studies indicate that the cost of a high-quality meal can be the main barrier and a method to lower the cost while maintaining the quality of meals is needed [37,38,39]. Although Vietnam’s Ministry of Training and Education and the Vietnamese National Institute of Nutrition have introduced a high-quality meal planning tool, it is unsuitable for many schools due to its high-cost requirement [5]. There is a disagreement between schools (or food providers) and parents. While principals and food suppliers supposed that better quality school meals for students would require a more considerable financial contribution from parents, parents were not willing to pay more [5]. However, our methodology successfully kept the meal costs while providing various ingredients. In our study, despite the increase in vegetables from 1.8 ± 0.2 in the current menus to 5.6 ± 0.9 in the new menus, the cost was almost the same at around 30,000 VND (1.2 USD). In fact, during the interviews, the principals thought that improving the meals meant they had to increase the portion size or total amount of vegetables, none of them had thought about dividing the standard 100 g into smaller quantities of more vegetable types. Our study highlights this method’s potential in improving school lunches and solving the meal cost problem.

The workload of kitchen staff was also taken into consideration in this study. We identified that the suppliers provided cut vegetables when making the new menus. So, with new menus, we ordered these cut vegetables and evaluated the working time and workload between the two menu types. There was no difference between the working time and workload of the current and new menus. Moreover, there is an upward trend in using industrial meal suppliers in Vietnamese primary schools, with an increase from 10% in 2011 [1] to about 66% in 2022 [3]. The cost per meal has to be paid by the companies and is the same as the cost of materials, which includes paid for workers. Hence, our methods could ensure that there is no overwork for kitchen staff, even those who work in school kitchens or catering companies. However, more evidence is needed in future studies to test the workload of school lunches and kitchen staff.

The major strength of this study was its cross-over design, which evaluated each subject’s response to both types of menus, excluding the individual preferences factors. The second strength of our study was that plate waste was measured by the weighing method, which is the most accurate evaluation method [40].

Nevertheless, this study has some limitations. First, our study was conducted at only one school. Thus, we would like to expand this method to more expansive areas in the future. Second, we used more root vegetables that are rich in soluble fiber compared to the same amount of leaf vegetables on current menus, which might lead to no difference in fiber intake of students as the Vietnamese food composition table contains insoluble fiber only [41]. Third, our newly developed menus are considered a novelty to the students, and the students might be aware that they are trying new foods. Last, a short-term intervention period allows for comparing intake amounts but not the application of the school in the follow-up weeks and the health effects on students. We recommend that future studies include a longer intervention time with more types of menus and consider the impact on the long-term physical health of children, such as body weight.

## 5. Conclusions

Diversifying combinations focusing on vegetables can significantly enhance both vegetable intake and meals’ sensory characteristics among students without increasing costs. This approach offers a scalable and practical strategy for schools to promote healthier eating habits among students.

## Figures and Tables

**Figure 1 nutrients-17-01385-f001:**
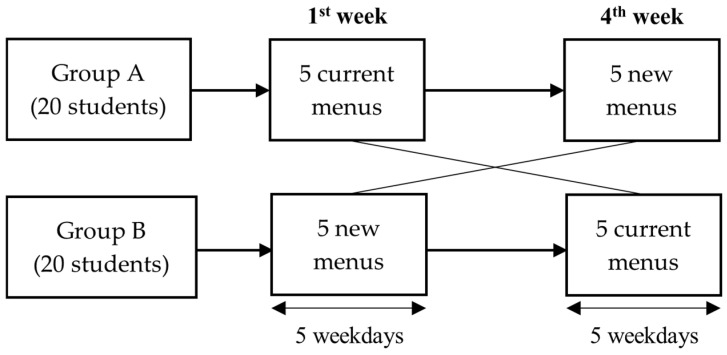
Study procedure.

**Figure 2 nutrients-17-01385-f002:**
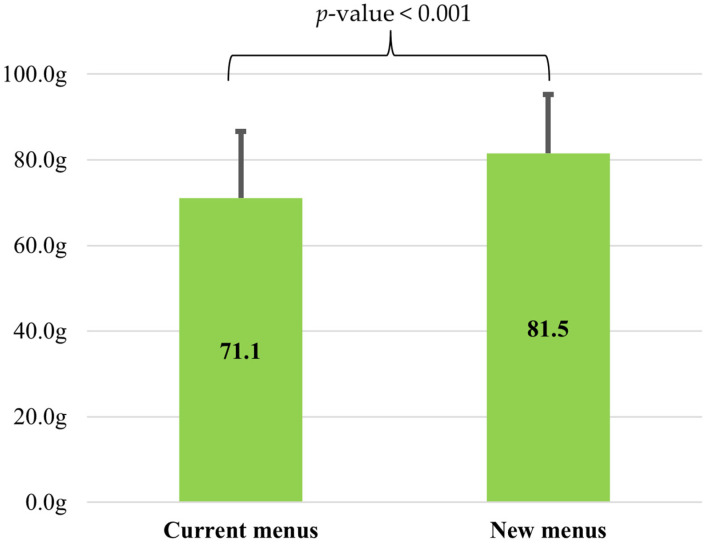
Mean vegetable intake of current and new menus (n = 40). Current and new menus provided the same as 100 g of vegetables per meal. Results are presented as mean ± SD. Student’s paired *t*-test was conducted between current and new menu values.

**Figure 3 nutrients-17-01385-f003:**
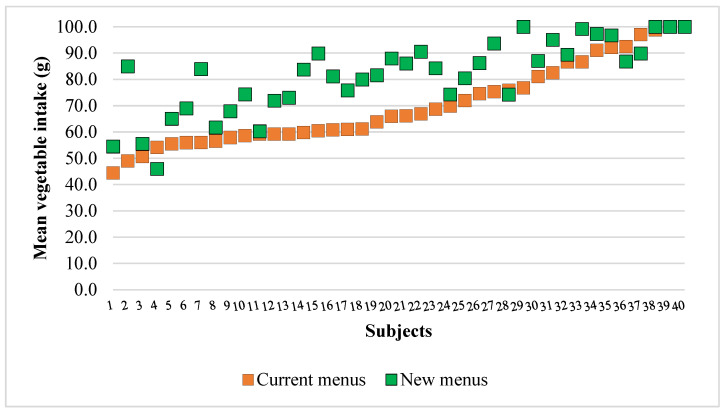
The individual differences in vegetable intake in current and new menus (n = 40). We sorted the results of current menus from the lowest to the highest. Results are presented as a mean of five current and five new menus for each subject. Student’s paired *t*-test was conducted between the current and new menu values.

**Table 1 nutrients-17-01385-t001:** Provision of current and new menus (n = 5).

	Current Menus	New Menus	*p*-Value
Cost (VND)	29,658 ± 1953	30,587 ± 1650	0.16
Food variety (number)	6.2 ± 0.8	10.2 ± 0.6	**<0.001**
Vegetable variety (number)	1.8 ± 0.2	5.6 ± 0.9	**<0.001**
Energy (kcal)	623 ± 35	630 ± 45	0.60
Protein (g)	28.0 ± 4.5	27.4 ± 4.1	0.57
Lipid (g)	19.5 ± 3.8	20.3 ± 3.8	0.13
Carbohydrate (g)	82.4 ± 1.8	82.9 ± 4.2	0.79

VND: Vietnamese currency (dong). Results are presented as mean ± SD. Students’ paired *t*-tests were conducted on the current and new menu values. Bold *p*-values show significant differences between current and new menus.

**Table 2 nutrients-17-01385-t002:** Sensory test scores of current and new menus (n = 40).

	Current Menus	New Menus	*p*-Value
Color	4.6 (4.2–4.9)	4.8 (4.3–5.0)	**0.02**
Smell	4.5 (4.0–4.8)	4.6 (4.2–5.0)	**0.02**
Taste	4.5 (4.2–4.8)	4.8 (4.4–5.0)	**0.01**
Texture	4.2 (3.8–4.5)	4.6 (4.0–5.0)	**0.02**
Overall	4.4 (4.2–4.8)	4.8 (4.4–5.0)	**0.01**

The score of each item ranges from 1 (dislike very much) to 5 (like very much). Results are presented as median values (25th–75th). Wilcoxon signed-rank test was conducted on the current and new menu values. Bold *p*-values show significant differences between current and new menus.

**Table 3 nutrients-17-01385-t003:** Energy and nutrient intakes in current and new menus (n = 40).

	Current Menus	New Menus	*p*-Value
Energy (kcal)	558 (514–596)	595 (556–619)	**<0.001**
Protein (g)	24.4 (21.7–26.1)	25.6 (23.4–26.9)	**0.03**
Lipid (g)	18.7 (17.9–19.4)	19.8 (19.2–20.2)	**<0.001**
Carbohydrate (g)	71.8 (65.6–78.6)	77.6 (70.2–81.1)	**<0.001**

Results are presented as median (25th–75th). Wilcoxon signed-rank test was conducted between the current and new menu values. Bold *p*-values significant difference between current and new menus.

## Data Availability

The original contributions presented in this study are included in the article. Further inquiries can be directed to the corresponding authors.

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
