# Peer review of "Enhancing Vietnamese Students’ Acceptance of School Lunches Through Food Combination: A Cross-Over Study"

_nutrients, 2025, doi:10.3390/nu17081385_

Round 1

Reviewer 1 Report

Comments and Suggestions for Authors

Line 46: What is meant by materials, different foods?
Line 48: 190 school meals in a year.
Line 49 instead of menus maybe “different meals then for Japan, while there were 190 different meals with 376….
Line 59” “the way they are” maybe this is clearer as “a simple dish with the vegetable as the main ingredient”
Line 73: consider changing “prove” to “test”
Line 207 to add to the idea of neophobia — it seems like the new menu not only increased vegetable intake overall and for 90% of the students but it also seemed to change (flatten) the trend line with those that were on the low end with the original menu being more similar in intake with those on the high end of the original menu. 
Line 281 another limitation may have been the novelty and the students being aware that they were trying new foods, more reasons for longer-term and larger studies.

Comments on the Quality of English Language

While I checked that it could use a review overall the English is good. In some of my initial comments in the introduction were places where I thought the English could be improved.

Author Response

Comments 1: Line 46: What is meant by materials, different foods?
Response 1: Thank you for your question. 
Materials here is meant “food materials”. Thus, to clarify the content of this sentence, it has been rewritten as below:
 “Although there is a regulation to have at least ten food materials in each meal, Vietnamese school meals do not meet this criterion [6,7].” (L46-47)

Comments 2: Line 48: 190 school meals in a year.
Response 2: Thank you for pointing this out. These words have been rewritten as below:
“…, both countries have 190 school meals in a year,…” (L48)

Comments 3: Line 49 instead of menus maybe “different meals then for Japan, while there were 190 different meals with 376….
Response 3: Thank you for your suggestion. We have fixed the sentence following your suggestion. The rewritten sentence is shown below:
“While Vietnam cycled 23 menus throughout 190 meals with 53 food ingredients, there were 190 different meals with 376 food ingredients in Japan.”(L49-L50) 

Comments 4: Line 59: “the way they are” maybe this is clearer as “a simple dish with the vegetable as the main ingredient”
Response 4: Thank you for your suggestion. We have changed the word phrase you pointed out. The rewritten sentence is shown below:
“Unlike the Japanese, Vietnamese cook all ingredients separately in a simple way, with all vegetables served as stir-fried or soup.” (L60-L61)

Comments 5: Line 73: consider changing “prove” to “test”
Response 5: Agree. We have, accordingly, modified “prove” to “test”.
The sentence after revised as below: 
“Based on the above, this study was conducted to test the hypothesis that food combinations in school lunch meals can positively enhance Vietnamese students’ acceptance of school lunches focusing on vegetables.” (L73)

Comments 6: Line 207 to add to the idea of neophobia — it seems like the new menu not only increased vegetable intake overall and for 90% of the students but it also seemed to change (flatten) the trend line with those that were on the low end with the original menu being more similar in intake with those on the high end of the original menu. 
Response 6: Thank you for your valuable suggestion to improve the manuscript. After revising this paragraph about neophobia, we admit that it contains an overextension of our study concern. Thus, we have decided to delete this paragraph from our manuscript to avoid confusion.

Comments 7: Line 281 another limitation may have been the novelty and the students being aware that they were trying new foods, more reasons for longer-term and larger studies.
Response 7: Thank you for pointing this out. We have included the limitation you suggested in L300-L301. The new sentence is shown below:
“Third, our newly developed menus are considered a novelty to the students, and the students might be aware that they are trying new foods.” (L320-L321)

Reviewer 2 Report

Comments and Suggestions for Authors

The authors present a crossover study examining the impact of serving a greater variety of vegetables in school lunch. The study features the gold standard plate waste measurement method and the study design facilitates causal inference. However, there are some limitations that need to be addressed prior to publication. My primary concerns are that the discussion contains statements that are not properly supported by references, more discussion of the relative change in consumption is needed (e.g. % consumed), and more information is needed about the sensory measures. Also, minor English language editing is needed. Additional comments below.

  1. What is the practical/clinical significance of the of a 10 gram increase in vegetable consumption? It would be helpful if the % of the vegetables consumed (amount eaten/amount served) could be compared before and after the intervention to better understand the real-world significance of the change. In addition, most school nutrition food waste studies express their results in these terms. Thus, this modification would facilitate other scientists citing your work.
  2. The sensory test scores for the current menu were already very high. This should be acknowledged.
  3. Please provide more information about the sensory tests used in this study. Were these measures newly developed the authors? Are these tests appropriate for children? In particular, how do the authors know that the children understand the concept of texture?
  4. Table 1: I suggest that the authors use the term food variety instead of food material and vegetable variety instead of vegetable material.
  5. Lines 181-182: Please provide a more balanced summary of the results by describing the lower range of vegetable intake along with the highest range (40 g increase).
  6. Line 196: Satisfaction was not assessed. Please re-phrase this finding to be more inline with what was actually measured.
  7. Line 216-218: Citation 18 does not contain any plate waste results. This sentence should be changed to state one study instead of several unless the authors can find better support for this statement.
  8. Lines 202-210: This paragraph seems to contain an overextension of the present studies results. The authors did not measure curiosity or motivations so the first sentence of this paragraph is inappropriate. Neophobia is very common in childhood. Not all children think trying new foods is interesting. This idea is poorly supported by the current reference (20), and more balanced explanation of what is known on neophobia/new foods is warranted. Further, the Sick et al (2019) study (ref 19) did not involve tasting or consumption of vegetables so this reference seems inappropriate here (line 209).
  9. Lines 230-234: Were the new vegetables high in glutamate? Were mushrooms included in the study? Did the days with highest consumption feature proteins high in inosinate and the days were lower consumption lack these substances? More information is needed to support the claim that this explained the findings of this study.
  10. Line 238: References 23 and 34 do not accurately support the statement that children prefer boiled vegetables. The Poelman study was conducted with 5-6 year olds, which are in a different stage of development than the children in the present study. Fried foods have several sensory principles that facilitate taste preferences.
Comments on the Quality of English Language

The quality of the English language could be improved.

Author Response

Comments 1: What is the practical/clinical significance of the of a 10 gram increase in vegetable consumption? It would be helpful if the % of the vegetables consumed (amount eaten/amount served) could be compared before and after the intervention to better understand the real-world significance of the change. In addition, most school nutrition food waste studies express their results in these terms. Thus, this modification would facilitate other scientists citing your work.
Response 1: Thank you for your question. We have added one paragraph about the value of increasing 10g of vegetables in the discussion. The added paragraph is shown below:
“Although a 10-gram increase in vegetable intake per meal may seem small, it is clinically meaningful for Vietnamese children aged 10–11, particularly in light of national concerns about both undernutrition and the rising prevalence of overweight and obesity [19]. Current data indicate that vegetable intake among Vietnamese school-aged children remains below recommended levels [20]. Therefore, small, achievable dietary changes, such as a 10-gram increase per meal, resulting in an additional 30 grams per day, are more likely to be adopted and sustained, especially in school settings. This incremental improvement can help move children's intake closer to the dietary guidelines set by the Vietnamese Ministry of Health and the National Institute of Nutrition.” (L213-L221)

Regarding the percentage of the vegetables consumed in comparison, we sincerely thank you for your valuable suggestion to enhance our study. As both the current and our newly developed menus served 100g of vegetables, the intake amount in grams can be understood as the percentage of vegetable intake. However, in Vietnam, studies usually indicate the quantity of vegetable intake in grams in comparison, and it is easy to compare with the Vietnamese government’s recommendation, which is shown as grams per day. Thus, we would like to show it as grams for further comparisons and for the government's consideration in improving the school lunch policy in Vietnam.

Comments 2: The sensory test scores for the current menu were already very high. This should be acknowledged.
Response 2: Thank you for your comments. We have modified the sentence for clearer. The revised sentence is shown as below: 
“Although the sensory test scores for the current menus were already high, they could be statistically improved with the new menus (p<0.05).” (L167-L168)

Comments 3: Please provide more information about the sensory tests used in this study. Were these measures newly developed the authors? Are these tests appropriate for children? In particular, how do the authors know that the children understand the concept of texture?
Response 3: Thank you for your comments. We have provided more information about the sensory tests in our manuscript, which is shown below. 
“Sensory tests were conducted using a 5-point Hedonic scale, following the questionnaire of author Nguyen Van Diep, with these five questions below [7]. 
•    How do you rate the color of this meal?
•    How do you rate the taste of this meal?
•    How do you rate the smell of this meal?
•    How do you rate the texture (uniform surface, crunchiness) of this meal?
•    Overall, how do you rate this meal?
Students then gave scores of 1 (dislike very much), 2 (dislike), 3 (neither dislike nor like), 4 (like), and 5 (like very much) for each question.” (L129-L137)

Comments 4: Table 1: I suggest that the authors use the term food variety instead of food material and vegetable variety instead of vegetable material.
Response 4: Thank you for your suggestion. We have changed the terms “food material” and “vegetable material” into “food variety” and “vegetable variety” in Table 1.

Comments 5: Lines 181-182: Please provide a more balanced summary of the results by describing the lower range of vegetable intake along with the highest range (40 g increase).
Response 5: Thank you for your comment. We have summarized more balance about Figure 3 result. The revised paragraph is shown below:
“Figure 3 illustrates that the majority of subjects – approximately 90% – consumed more vegetables with the new menus compared to the current ones. The increase in vegetable intake varied among subjects. While the most substantial improvement reached up to 40g, indicating a significant dietary change for some individuals, others showed only modest increases or no change at all. A few subjects even showed minimal or no improvement in their intake. Notably, subjects with initially low consumption on the current menus were able to boost their intake on the new menus significantly. Additionally, more subjects consumed the whole vegetable portion under the new menus, in contrast to just one subject who did so under the current menu. These findings suggest that individual responses varied while the new menus were broadly effective.” (L185-L194)

Comments 6: Line 196: Satisfaction was not assessed. Please re-phrase this finding to be more inline with what was actually measured.
Response 6: Thank you for pointing this out. We have rephrased the sentence to align with the study’s results. The revised sentence is shown below:
“Based on the sensory test scores, our study suggests that combining food materials had a positive impact on the meals’ sensory qualities.” (L222-L223)

Comments 7: Line 216-218: Citation 18 does not contain any plate waste results. This sentence should be changed to state one study instead of several unless the authors can find better support for this statement.
Response 7: Thank you for pointing this out to us. We have modified this paragraph as below:
“Based on the sensory test scores, our study suggests that combining food materials had a positive impact on the meals’ sensory qualities. Attributes such as appearance, smell, and texture are important predictors of whether children choose to eat a meal [21-23]. Francesca Zampollo emphasized that children tend to prefer plates that include a variety of items and vibrant colors [24]. Similarly, Nguyen Van Diep reported a 24.3g increase in vegetable intake among students when meals were enhanced in sensory test scores, specifically in terms of color, smell, taste, and texture [7]. Aligned with these findings, our results showed that the new menus, which scored higher in color, smell, and texture, were associated with lower quantities of vegetables left uneaten compared to the current menus.” (L222-L231)

Comments 8: Lines 202-210: This paragraph seems to contain an overextension of the present studies results. The authors did not measure curiosity or motivations so the first sentence of this paragraph is inappropriate. Neophobia is very common in childhood. Not all children think trying new foods is interesting. This idea is poorly supported by the current reference (20), and more balanced explanation of what is known on neophobia/new foods is warranted. Further, the Sick et al (2019) study (ref 19) did not involve tasting or consumption of vegetables so this reference seems inappropriate here (line 209).
Response 8: Thank you for your valuable comments on improving the manuscript. Since we try to make the most familiar meals to subjects’ eating habits and appropriate to Vietnamese food culture, after revising this paragraph about neophobia among children, we admit that this contains an overextension of our study concern. Thus, we have deleted this paragraph from our manuscript to avoid confusion.

Comments 9: Lines 230-234: Were the new vegetables high in glutamate? Were mushrooms included in the study? Did the days with highest consumption feature proteins high in inosinate and the days were lower consumption lack these substances? More information is needed to support the claim that this explained the findings of this study.
Response 9: Thank you for your valuable comment. We have added a paragraph to give more information to support our explanation. The added paragraph is shown below:
“In our study, several high-glutamate vegetables permitted in the materials list, such as tomatoes, and cabbage [34], facilitated the development of menus leveraging the synergistic taste effect. Observational data indicated that students responded most positively to the meal served on Wednesday (Table S1). In this menu, the original ketchup to eat with a fish ball was replaced with a tomato-based sauce containing diced carrots and onions, creating a combination of glutamate (tomato, carrot, and onion) and inosinate (fish) that appeared to enhance palatability among participants. Similarly, on Friday, the mixed rice dish (comprising rice, vegetables, egg, and pork) and vegetable-only side dish (cabbage, tomato, and cucumber) were offered (Table S1). Although no statistical analysis was conducted, anecdotal observations suggested that students were more likely to consume vegetables when incorporated into the mixed rice (including protein-based ingredient), while vegetable-only dishes were left uneaten with larger amounts. Unfortunately, mush-rooms were not included in the study. According to the vice principal, purchasing mushrooms in small quantities was deemed cost-prohibitive, potentially increasing the overall cost of the new menus. Therefore, future research is recommended to include a broader sample of subjects to examine the effects of incorporating mushrooms into combinations of vegetables on vegetable consumption and menu acceptability.” (L246-L262)

Comments 10: Line 238: References 23 and 34 do not accurately support the statement that children prefer boiled vegetables. The Poelman study was conducted with 5-6 year olds, which are in a different stage of development than the children in the present study. Fried foods have several sensory principles that facilitate taste preferences.
Response 10: Thank you for your valuable comments. We have excluded reference 23 from our manuscript owing to their subject's age. We agree that fried foods facilitate taste preferences. However, reference 34 pointed out, “the undesirable brown colouring on the stir-fried and deep-fried vegetables did not favour liking”. To clarify how reference 34 supports our study, we have rewritten these sentences in the manuscript. 
“The preparation method also influenced vegetable consumption among children [35]. Research specifically examining the vegetable preparation preferences of children 10 to 11 is limited. However, Zeinstra et al. (2010) concluded that all age groups in their study, including 11-12-year-old children, preferred steamed and boiled vegetables over mashed, stir-fried, grilled, and deep-fried, with the most familiar preparation belonging to boil. Crunchiness and color also proved to be characteristics of vegetable preferences among children. Thus, shorter boiling times for vegetables might result in retaining crunchiness and avoid undesirable brown coloring on the stir-fried, enhancing children’s willingness to finish their vegetable dishes. Regardless of the above, vegetable cooking methods are limited in Vietnamese school meals.” (L263-L271)

Round 2

Reviewer 2 Report

Comments and Suggestions for Authors

The authors have done an excellent job responding to my comments. I think the new paragraph in the discussion (L245-261) strengthens the replicability of the study and addresses a gap in what has currently been published about school food plate waste.

Comments on the Quality of English Language

The only remaining issue with the paper are English languages issues. For example:

Line 168: "Could" implies possibility. If there improvements occurred, could is inappropriate.

Lines 270-272: It is unclear if the authors are referring to Vietnamese policies/program standards, past research, or present study methods.

Also the term "materials" is not typically used in English to describe food in a culinary application. Other words like ingredients or menu might be appropriate. Using the term foods or vegetables may be more appropriate in some cases as well.

Author Response

3. Point-by-point response to Comments and Suggestions for Authors
Comments: The authors have done an excellent job responding to my comments. I think the new paragraph in the discussion (L245-261) strengthens the replicability of the study and addresses a gap in what has currently been published about school food plate waste.
Response: Thank you for your valuable suggestions and comments on how to improve our manuscript.

4. Response to Comments on the Quality of English Language
Point 1: Line 168: "Could" implies possibility. If there improvements occurred, could is inappropriate.
Response 1: Thank you for pointing this out. We have modified this sentence, which is shown below:
“Although the sensory test scores for the current menus were already high, they were statistically improved with the new menus (p<0.05).” (L168)

Point 2: Lines 270-272: It is unclear if the authors are referring to Vietnamese policies/program standards, past research, or present study methods.
Response 2: Thank you for pointing this out to clear our manuscript. We have fixed the sentence for clarity. The revised sentence is shown below:
“Regardless of the above, from our observation, vegetable cooking methods were limited in Vietnamese school meals. A previous study reported that all vegetables served in primary schools' lunches were stir-fried or included in soup, but no boiled vegetables or salads were served [5].”(L270-L271)

Point 3: Also the term "materials" is not typically used in English to describe food in a culinary application. Other words like ingredients or menu might be appropriate. Using the term foods or vegetables may be more appropriate in some cases as well.
Response 3: Thank you for your suggestion. We have modified the term “materials” in our manuscript. Detailed modifications:
“Although there is a regulation to have at least ten food materials in each meal,…” to 
“Although there is a regulation to have at least ten food ingredients in each meal,…” (L46)

“Although the cost was the same between current and new menus, food and vegetable materials in new menus increased 1.6 and 3.1 times compared to current menus (p<0.001).” to 
“Although the cost was the same between current and new menus, food and vegetable varieties in new menus increased 1.6 and 3.1 times compared to current menus (p<0.001).” (L160-L162)

“Based on the sensory test scores, our study suggests that combining food materials had a positive impact on the meals’ sensory qualities.” to 
“Based on the sensory test scores, our study suggests that combining food ingredients had a positive impact on the meals’ sensory qualities.” (L222-L223)

“In our study, several high-glutamate vegetables permitted in the materials list,…” to 
“In our study, several high-glutamate vegetables permitted in the food ingredients list,…” (L245)

“However, our methodology successfully kept the meal costs while providing various food materials. In our study, despite the increase in vegetable materials from 1.8 ± 0.2 in the current menus to 5.6 ± 0.9 in the new menus,…” to 
“However, our methodology successfully kept the meal costs while providing various ingredients. In our study, despite the increase in vegetables from 1.8 ± 0.2 in the current menus to 5.6 ± 0.9 in the new menus,…” (L292-L294)

We think the term “materials” is fine to keep in this sentence: “The cost per meal has to be paid by the companies and is the same as the cost of materials, which includes paid-for workers.” (L307-LL308)
